# Atlantic water flow through the Faroese Channels

Bogi Hansen[1], Turið Poulsen[1], Karin Margretha Húsgarð Larsen[1], Hjálmar Hátún[1], Svein Østerhus[2], Elin Darelius[3], Barbara Berx[4], Detlef Quadfasel[5], Kerstin Jochumsen[5]

[1]Faroe Marine Research Institute, PO Box 3051, FO-110 Tórshavn, Faroe Islands
[2]Uni Research Climate and the Bjerknes Centre for Climate Research, Nygårdsgata 112, N-5008 Bergen, Norway
[3]Geophysical Institute, University of Bergen and the Bjerknes Centre for Climate Research, Allég. 70, 5007 Bergen, Norway
[4]Marine Scotland Science, 375 Victoria Road, Aberdeen, AB11 9DB, UK
[5]Institut für Meereskunde, Universität Hamburg, Bundesstrasse 53, 20146 Hamburg, Germany

*Correspondence to*: Bogi Hansen (bogihan@hav.fo)

**Abstract.** Through the Faroese Channels – the collective name for a system of channels linking the Faroe-Shetland Channel, Wyville Thomson Basin and Faroe Bank Channel – there is a deep flow of cold waters from Arctic regions that exit the system as overflow through the Faroe Bank Channel and across the Wyville Thomson Ridge. The upper layers, in contrast, are dominated by warm, saline water masses from the southwest, termed Atlantic water. In spite of intensive research over more than a century, there are still open questions on the passage of these waters through the system with conflicting views in recent literature. Of special note is the suggestion that there is a flow of Atlantic water from the Faroe-Shetland Channel through the Faroe Bank Channel, which circles the Faroes over the slope region in a clockwise direction. Here, we combine the observational evidence from ship-borne hydrography, moored current measurements, surface drifter tracks, and satellite altimetry to address these questions and propose a general scheme for the Atlantic water flow through this channel system. We find no evidence for a continuous flow of Atlantic water from the Faroe-Shetland Channel to the Faroe Bank Channel over the Faroese slope. Rather, the southwestward flowing water over the Faroese slope of the Faroe-Shetland Channel is totally re-circulated within the combined area of the Faroe-Shetland Channel and Wyville Thomson Basin, except possibly for a small release in the form of eddies. This does not exclude a possible westward flow over the southern tip of the Faroe Shelf, but even including that, we estimate that the average volume transport of a "Circum-Faroe Current" does not exceed 0.5 Sv (1 Sv = $10^6$ m$^3$ s$^{-1}$). Also, there seems to be a persistent flow of Atlantic water from the western part of the Faroe Bank Channel into the Faroe-Shetland Channel that joins the Slope Current over the Scottish slope. These conclusions will affect potential impacts from offshore activities in the region and they imply that recently published observational estimates of the transport of warm water towards the Arctic obtained by different methods are incompatible.

## 1 Introduction

The Faroese Channels (Fig. 1) are the deepest passage across the submarine ridge system between Iceland and the Scottish continental shelf and among the key locations for exchange between the Nordic Seas and the rest of the World Ocean. The

name seems to have been coined by Dooley and Meincke (1981) for the Faroe-Shetland Channel and its continuation into the Faroe Bank Channel. This includes the basin east of the Wyville Thomson Ridge, the Wyville Thomson Basin.

There is a continuous flow of cold, dense water of Arctic origin flowing through the deep parts of this channel system (Fig. 1) that leaves the Norwegian Sea through the Faroe Bank Channel (Hansen and Østerhus, 2007) and as a weak overflow across the Wyville Thomson Ridge (Sherwin et al., 2008). The upper layers, in contrast, are dominated by warm and saline waters from the West, which we term the Atlantic inflow to the Nordic Seas (Hansen and Østerhus, 2000).

The Atlantic inflow has a weak branch west of Iceland, but most of it enters the Norwegian Sea between Iceland and the European continent (Østerhus et al., 2005). As we move from the northern parts of the inflow region towards the Scottish Shelf, the water increases in both temperature and salinity, but most of the upper water over the open ocean has traditionally been considered one water mass, termed Modified North Atlantic Water (MNAW). Over the Scottish slope, considerably warmer and more saline water, termed North Atlantic Water (NAW), is carried by the Slope Current (e.g.Booth and Ellett, 1983).

As the Atlantic water approaches from the open Northwest Atlantic, it splits into two branches with one branch flowing north of the Faroes and one branch flowing into the Faroe-Shetland Channel. After the northern branch has crossed the Iceland-Faroe Ridge, it gets focused into a narrow boundary current, the Faroe Current (Hansen et al., 2003, 2015). The southern branch of the Atlantic water flows into the Wyville Thomson Basin and continues into the Faroe-Shetland Channel where it joins the Slope Current to flow northeastwards through the Faroe-Shetland Channel with the core over the slope on the Scottish side of the channel (Booth and Ellett, 1983).

Over the slope on the Faroese side of the Faroe-Shetland Channel, in contrast, the typical flow is towards the Southwest, at least in the northeastern part of the Faroe-Shetland Channel. This current was shown already by Helland-Hansen and Nansen (1909; their Fig. 29) as a part of the Faroe Current that is retroflected into the Faroe-Shetland Channel. Hátún (2004) called this current the Southern Faroe Current (SFC; Fig. 1). In the surface layers, the waters of the Southern Faroe Current are only slightly colder and less saline than the original Atlantic water feeding the Faroe Current and the Atlantic layer in the Faroe Bank Channel and Wyville Thomson Basin (Larsen et al., 2012). Due to mixing with water of Arctic origin north of the Faroes, the water in the Southern Faroe Current, however, decreases in both temperature and salinity much faster with depth (Becker and Hansen, 1998). In the deeper parts of the Atlantic layer, we therefore find large differences between the Southern Faroe Current and the originally inflowing Atlantic water. Where the two water masses meet, we find a front, which in temperature and salinity is most pronounced in the deep parts of the Atlantic layer and was called the "Midwater Front" by Hansen and Jákupsstovu (1992).

This front is indicated already in the original work by Helland-Hansen and Nansen (their Fig. 22) and seems to have inspired them to suggest a complete re-circulation of the Southern Faroe Current within the Faroe-Shetland Channel (their Fig. 29). Based on results from the ICES (International Council for the Exploration of the Sea) Overflow-73 experiment, Dooley and Meincke (1981), on the other hand, suggested that the Southern Faroe Current splits and that approximately half of it (1.2 Sv, 1 Sv = $10^6$ m$^3$ s$^{-1}$) continues through the Faroe Bank Channel. This view was challenged by Hansen and

Jákupsstovu (1992) and by Becker and Hansen (1998), mainly based on water mass properties. In their interpretation, there is a more or less closed clockwise circulation around the Faroes on the shelf (white arrows in Fig. 1), but not off the shelf.

This scheme, with Southern Faroe Current re-circulation within the Faroe-Shetland Channel, seems to have been widely adopted (e.g., Poulain et al., 1996; Turrell et al., 1999; Sherwin et al., 1999; 2006; Hansen and Østerhus, 2000; Hunegnaw et al., 2009; Chafik, 2012). Recently, however, it has been questioned by Rossby and Flagg (2012), who returned to the scheme of Dooley and Meincke (1981) with a splitting of the Southern Faroe Current suggesting that 1 Sv of water from the Southern Faroe Current continues into the Faroe Bank Channel. Rossby and Flagg (2012), furthermore, suggest that this flow is part of a "tidally-driven Circum-Faroe Current", which circulates the Faroes off the shelf. To distinguish this current from the well established circulation on the Faroe Shelf (Larsen et al., 2008), we will use the name "Circum-Faroe Slope Current", abbreviated to CFSC (dashed arrow in Fig. 1).

The existence or non-existence of a CFSC is the main question addressed in this study and, a priori, it might seem of little relevance. It is, however, critical for assessments of potential environmental impacts from offshore oil and gas activity in the Faroe-Shetland Channel and it may also be critical for quantifying the transport of warm water and heat towards the Arctic. Even in recent literature, there are considerable differences between transport values published in different studies and the existence/non-existence of a CFSC is found to play a critical role.

Historically, the arguments against a CFSC have mainly been based on hydrography, whereas Dooley and Meincke (1981) as well as Rossby and Flagg (2012) used direct current measurements to argue for its existence. These measurements were, however, not made in the critical boundary region between the Faroe-Shetland Channel and the Faroe Bank Channel. Here, we present results from moored current measurements in this region, which are combined with data from satellite altimetry, hydrography, and surface drifters. The methodology is described in brief, but where relevant, full details are found in the supplementary material.

## 2 Material and methods

Locations of the main observational sites are shown in Fig. 2.

### 2.1 In situ current measurements

We use current measurements from a number of moored instruments. In shallow waters (filled red circles in Fig. 2a), these have mainly been single-point current meters (Aanderaa) at 40 m depth (Table S1 in the supplement). In waters deeper than 200 m, most of the current measurements have been acquired by upward-looking ADCPs (Acoustic Doppler Current Profilers, open red circles on Fig. 2) that have been deployed either at depths below the main fishing activity or in trawl-protected frames (Table S2). The mooring sites are labeled using two uppercase letters, where the first letter indicates the section along which the mooring is located: the F-section, the Z-section, the S-section, or the E-section (Fig. 2). We use de-tided data averaged to daily values.

## 2.2 Hydrography

Most of our hydrographic data were collected by R/V Jens Christian Svabo or R/V Magnus Heinason between 1976 and 2015. These data consist of quality controlled CTD (Conductivity, Temperature, Depth) profiles distributed over the whole region, but more regularly on three standard sections, the V-section, the S-section, and the E-section (Fig. 2a). In addition, we use CTD data along the Z-section (Fig. 2a, c) from two cruises by MRV Scotia. Different instrument models and different calibration procedures have been used. Thus, the CTD data have varying quality, with the most reliable salinity values after 1995. In every case, the data quality is, however, more than sufficient for our requirements.

## 2.3 Altimetry

Altimetry data were downloaded from the global gridded (0.25°x0.25°) AVISO+ data set available from "http://www.aviso.altimetry.fr". We downloaded both the *Mean Dynamic Topography* (Fig. S1) and daily *Sea Level Anomalies (SLA)* from 1 January 1993 to 10 September 2015. Adding the values from the two data sets gives daily values of *Absolute Dynamic Topography*. Much of the variation in sea level in the region involves the whole region moving up or down, which in some cases may introduce unnecessarily noisy data. To reduce the noise level, we therefore generated a new data set by subtracting for each time step the area-average for the region 58.5–64° N by 2–10° W from the value in each grid point. For some applications (e.g., Fig. S7), these data give a better representation of the local dynamics, and we denote them *SLA\**.

Altimetry points used in this study are labeled by a lowercase letter (f or z) followed by a number and are indicated on Fig. 2a by filled green circles. They are located along two sections, termed the f-section and the z-section, intended to follow two of the ADCP sections, the Z-section and the F-section, respectively.

## 2.4 Surface drifters

Data on surface drifter tracks for the region were downloaded from the NOAA AOML website (http://www.aoml.noaa.gov/envids/gld/dirkrig/parttrk_spatial_temporal.php). We selected all drifters that entered specified areas in our region, 104 drifters in total. The drifters covered the period from 1991 to 2016 and had drogues at 15 m depth. Drifter tracks for which the drogue had been lost were not included.

## 2.5 Statistical methods

Throughout the manuscript, we use standard linear (Pearson) correlation and regression analysis. In order to assess the statistical significance of the correlation coefficients, we have used the "Modified Chelton method" recommended by Pyper and Peterman (1998) to correct for serial correlation in the data. Significance level is indicated by asterisks attached to the correlation coefficient. * indicates $p < 0.05$, ** indicates $p < 0.01$, *** indicates $p < 0.001$. All of these are two-tailed

probabilities. The uncertainty intervals of regression coefficients presented are 95% confidence intervals similarly calculated using the degrees of freedom determined by the Modified Chelton Method.

## 3 Results

In this section, we present the results from the different observational data sets individually, but combine them where appropriate.

### 3.1 Current measurements from moored instruments

The results from the moored current measurements are used to give an overview of the Atlantic water large-scale circulation and to see in more detail how this water passes through the two ADCP sections: the Z-section and the F-section (Fig. 2).

#### 3.1.1 Large-scale Atlantic water circulation

An overview of the Atlantic water flow through the Faroese channels, based on moored current meter measurements, is presented by the red arrows in Fig. 3. Unfortunately, the ADCP measurements do not reach all the way to the surface, but they still reach well above the average depth of the deep boundary of the Atlantic water (Fig. 2b, c), which may be defined by the 5 °C isotherm (Berx et al., 2013). We have typically selected depths around 200 m to represent the Atlantic layer over deeper regions. At some sites, the velocity vectors in Fig. 3 are based on several deployments and should be good long-term representations. At other sites, only one or two deployments were available (Table S1 and Table S2).

In the Faroe-Shetland Channel, our main focus is on the Faroese side of the channel, but we have included a few deployments on the Scottish side. They illustrate the established (e.g., Berx et al., 2013) warm, saline Atlantic inflow from the West that flows over the Scottish slope and continues northeastwards into the Norwegian Sea.

On the Faroese side of the Faroe-Shetland Channel, Atlantic water - deriving from the inflow north of the Faroes - enters the channel at its northeastern end and flows towards the Southwest as the Southern Faroe Current. This current is focused over the Faroe slope and clearly evident at site EB and site SX (Fig. 3). We do not, however, find significant correlations in the along-channel velocity records obtained from the different mooring arrays across the slope, neither within the Faroe-Shetland Channel (e.g. EB – SB) nor between the Faroe-Shetland Channel and the Faroe Bank Channel (e.g. SB – FG, Table S3).

#### 3.1.2 Atlantic water flow through the Z-section

The Z-section is located on the border between the Faroe-Shetland Channel and the Wyville Thomson Basin and well suited to illustrate the exchange between these two regions. From Fig. 3, the average flow at 200 m depth during two deployments was towards the East across the whole off-shelf part of the section, except for site ZA. This site is where we would expect a possible continuation of the Southern Faroe Current to cross from the Faroe-Shetland Channel to the Wyville Thomson

Basin, but the average velocity at 200 m depth (averaged over 684 days) was less than 1 cm s$^{-1}$, which is the reason why no arrow is shown for this site in Fig. 3.

A time series plot of eastward velocities at ≈200 m depth through the Z-section (Fig. 4) shows that there are periods, when Atlantic water flows westwards through the Faroese side of the Z-section, especially at ZA. From Fig. 4, it appears that the velocities on the Faroese side (ZA and ZB) of the channel co-vary, as do the velocities on the Scottish side (ZC and ZE), whereas the two sides are in anti-phase. This is verified by a correlation analysis (Table 1), which also shows that the velocity at the shallower ZQ site (Fig. 2) was not significantly correlated with the velocity at ZA.

The negative correlation between the flows over the two slopes on the Z-section may be interpreted in two different ways. In one interpretation, we can assume a fairly constant unidirectional (eastward) flow through the Z-section that shifts between the two sides of the channel. The alternative interpretation involves re-circulation. In this interpretation, much of the water that in some periods flows westward over the Faroese side, is re-circulated farther west and joins the eastward flow over the Scottish side, intensifying it.

Using the velocity at 200 m depth at ZA as representative for Atlantic water flow ignores the fact that the depth of the Atlantic layer varies in time. To some extent, this variation may be seen in the bottom temperature at the site, measured by the ADCP, as shown by the bottom panels in Fig. 4. This figure indicates that westward flow at ZA is associated with lower bottom temperatures, i.e. a shallower Atlantic layer, than eastward flow. This is confirmed by splitting the ADCP velocity profiles into days warmer than and colder than 5 °C (Fig. S2)

The average velocity profiles at ZA (Fig. S2) do not appear to be very barotropic, but more relevant when we wish to link ADCP data and altimetry data (Sect. 3.2) is the vertical structure of the velocity variations. For the 2013–2014 deployment, the ADCP at ZA managed to profile persistently up to a depth of 57 m. When comparing the eastward velocities at this depth and at 257 m depth, we found a high correlation coefficient and a regression coefficient fairly close to 1 (Table 2). Similar results are found for the 2011–2012 deployment at ZA. Thus, the velocity variations are fairly barotropic within the Atlantic layer at ZA, which implies that variations in sea level slope at this location should be tightly coupled to the velocity variations throughout the Atlantic layer and hence also to transport..

### 3.1.3 Atlantic water flow through the F-section

The two long-term ADCP sites on the F-section, FB and FC, are in the deep part of the channel with the remaining site, FG, over the Faroese slope. At FB and FC, the ADCPs have been so deep that they do not reach close to the surface. For each of the three ADCP sites in the Faroe Bank Channel, we chose a reference depth, for which only a few days were error-flagged, but still is typically within the Atlantic layer (Fig. 2b). Correlation coefficients between pairs of weekly averaged along-channel velocities at these depths were positive and significant (Table 3). Although average velocities at both sides of the channel (FC and FG) are in opposite directions (Fig. 3), this implies that temporal variations are in phase across the channel in contrast to the Z-section.

Choosing those deployments of long duration that reached the shallowest levels for FC and FB, we can check how barotropic the variations in along-channel velocity are at these sites, as well as at FG. For FC and FG, we again find high correlations and regression coefficients close to 1 (Table 2) indicating barotropic velocity variations. For FB, the correlation coefficient was much lower and not statistically significant. This may perhaps be due to the location of FB close to the boundary between the oppositely flowing layers at both sides of the channel (Fig. 3).

## 3.2 Surface currents from altimetry

In this sub-section, we compare altimetry data with our ADCP measurements to check how well the horizontal variations of SLA values reflect current velocity variations within the Atlantic layer. The results are then used to generate long time series of velocity.

To investigate how well the altimetry data represent the flow of Atlantic water through the Z-section, we compare ADCP velocities on this section and anomalies of sea level tilt derived from the SLA values of altimetry points z1 - z6 (Fig. 2). We focus especially on the flow past ADCP site ZA, because this is where we expect the core of westward flow to be (Fig. 3 and Fig. 4). For all the cases in Table 4, the correlation coefficient increases substantially when going from daily to weekly averaged data, as expected, since this should increase the signal to noise ratio in the altimetry data and also allow geostrophy better time to establish balance.

For the two deployments at site ZA, we find highly significant correlation coefficients between along-channel velocity at ≈200 m depth and SLA difference between pairs of altimetry points (Table 4). Checking various combinations of altimetry points, we find that the best correlation is with the SLA difference just north of the site (z3–z2; Fig. 2a). That we find such high correlations even 200 m below the surface is consistent with our finding that the velocity variations at ZA are highly barotropic (Table 2). This can be utilized to reproduce the velocity at ZA for the whole altimetry period by a regression analysis:

$$U_{ZA}(t) = \alpha \cdot \Delta h_{23}(t) + \beta \qquad (1)$$

where $U_{ZA}(t)$ is the eastward velocity at ≈200 m depth at ZA, $\Delta h_{23}(t)$ is the SLA difference between z3 and z2, and $\alpha$ and $\beta$ are regression coefficients.

Motivated by the high correlation, we have reproduced a time series of the eastward velocity at 200 m depth at ZA for the period 1 Jan 1993 to 31 Dec 2014 using Eq. (1) with values of $\alpha$ and $\beta$ from Table 4. The blue curve in Fig. 5 shows annually filtered reproduced velocities. Clearly, there are periods, when the westward velocity at this location approaches 10 cm s[-1] even for a 13-month average, but other years have eastward net flow and the time series average remains small (1.2 cm s[-1] westward).

Going from ZA towards more central parts of the channel, to ZB and ZC (Fig. 2), the correlation coefficients with altimetry in Table 4 decrease, but then increase again as we move to ZE at the Scottish side of the channel especially for the second (and more prolonged) deployment at ZE. Due to range limitation, the ADCP only had complete data up to 268 m

depth during this deployment, but the difference in SLA was still highly correlated with the observed eastward velocity at this depth.

In the Faroe Bank Channel, the difference in SLA between f1 and f2 ($\Delta SLA_{FBC}$, Fig. 5) represents the along-channel surface velocity, horizontally averaged between the two points. While there is a strong gradient in the mean velocity across the Faroe Bank Channel (Fig. 3), the moorings across the channel co-vary (Table 3), and the variability is related to the difference in SLA between f1 and f2 (Table 4). Even though the ADCP reference depths for FC and FB are far below the surface, the correlation coefficients for these sites in Table 4 are fairly high and statistically significant especially for 15-day averaged data. This is consistent with the finding by Darelius et al. (2015) that the variability in the strength of the cold outflow below the Atlantic layer correlates with the sea surface slope. While the variability in $\Delta SLA_{FBC}$, thus, to some extent represents the variability in the Atlantic water flow through the Faroe Bank Channel there is no relationship between $\Delta SLA_{FBC}$ and the flow past ZA (Fig. 5).

### 3.3 Hydrography

We use the hydrographic data set to illustrate the coupling between circulation and water mass distribution on the Z-section and to illustrate the water mass difference between the Faroe-Shetland Channel and the Faroe Bank Channel.

### 3.3.1 Water mass distribution on the Z-section

We do not have many cruises with good coverage of both CTD and ADCP data on the Z-section simultaneously, but Fig. 6 shows data from two MRV Scotia cruises, during which ADCPs were deployed at ZA, ZB, and ZC. These two cruises were during different circulation states, which is reflected in the hydrography.

During the first cruise, in October 2011, water warmer than 9 °C and more saline than 35.30 (gray on Fig. 6c, e) dominated the upper 300–400 m layer all across the deep parts of the channel. There was a high-salinity core over the Scottish slope, but otherwise, this water mass was fairly homogeneous and flowed eastwards from the Wyville Thomson Basin into the Faroe-Shetland Channel as a broad, sluggish flow, seen at all three ADCP sites. Consistent with the sea level slope (Fig. 6a), east-going Atlantic inflow from the West seems to have dominated the upper layer from the Faroese shelf edge onto the Scottish shelf.

During the second cruise, in contrast, the homogeneous water mass was more confined to the Scottish side of the channel and the warm, saline upper layer on the Faroese side was much shallower. This layer flowed westwards at ZA with speeds around 30 cm s$^{-1}$, but water with the same properties flowed eastwards into the Faroe-Shetland Channel at ZC (Fig. 6h), consistent with the sea level slope (Fig. 6b). Thus, the hydrographic conditions during this cruise are consistent with a re-circulation scheme where water from the Southern Faroe Current flowed from the Faroe-Shetland Channel into the Wyville Thomson Basin over the Faroese slope, but returned to the Faroe-Shetland Channel over the Scottish side of the channel, partly at greater depth.

### 3.3.2 Water mass differences between the Faroe-Shetland Channel and the Faroe Bank Channel

The main argument for Becker and Hansen (1998) to reject a splitting of the Southern Faroe Current and the existence of a CFSC (Fig. 1) was based on the differences in water mass properties in the Faroe-Shetland Channel and Faroe Bank Channel (Fig. S3). The argument has been discussed by Hansen and Østerhus (2000) and will not be repeated here although we may note that the updated hydrographic data set still maintains this difference (Fig. S4).

The border between the Southern Faroe Current and the Atlantic inflow from the West is the Midwater Front that has a variable location (Hansen and Jákupsstovu, 1992). A key difference between the two water masses on either side of this border is that the temperature decreases much faster with depth in the Southern Faroe Current compared to the Atlantic inflow (e.g., Fig. 20 in Hansen and Østerhus, 2000).

The instantaneous location of the Midwater Front may therefore be mapped by considering the temperature in midwater, e.g. at 300 m depth. This is done in Fig. 7 where approximate extreme eastern (Fig. 7a), and western (Fig. 7b) locations for the Midwater Front are indicated by the black curves. These extreme locations have been drawn from the distributions of red and blue dots. The extreme eastern location excludes some red dots in the northeastern part of the Faroe-Shetland Channel that may be meanders of warm water from the Slope Current (Sherwin et al., 2006; Chafik, 2012). The extreme western location, likewise, excludes a few blue dots that may be eddies. The two extreme locations, therefore, are subjectively drawn and should not be interpreted too accurately. Nevertheless, this should give an indication of the area, within which the Midwater Front is typically found.

### 3.4 Surface drifter passage through the Faroese Channels

With a sufficient number passing through the region, the tracks of surface drifters should give an impression of the general surface circulation and indicate linkages between the various parts of the Faroese channels.

In Fig. 8, we have plotted tracks of all surface drifters that entered the region from the Southwest, or from the Faroe Current (defined by boxes in Fig. 8). None of the 16 drifters from the Southwest (Fig. 8a) turned northwestwards through the Faroe Bank Channel. One of them (brown) crossed the Faroe Bank Channel from the shallow parts of Faroe Bank onto the shallow parts of the Faroe Plateau. Another drifter (blue) crossed the Faroe-Shetland Channel more than once. All the other drifters from the Southwest (red) flowed over the slope or shelf on the Scottish side of the Faroe-Shetland Channel.

A total of 89 drifters entered the area north of the Faroes (Fig. 8b), most of them following the core of the Faroe Current. Several of these turned southwards into the Faroe-Shetland Channel, but most of them re-circulated and turned northeastwards close to the eastern entrance to the Faroe-Shetland Channel. Only one drifter (blue) passed from that entrance into the Wyville Thomson Basin where it re-circulated and joined the northeastward flow through the Faroe-Shetland Channel, although crossing the channel several times. A few drifters in Fig. 8b entered the Faroe Bank Channel, but none of them came from the Faroe-Shetland Channel. Detailed tracks of all the drifters in the data set that passed through the Faroe Bank Channel (Fig. S5) confirm that they all came from the West.

**4 Discussion**

In addition to the in situ data, this study relies substantially on altimetry, which is known to have limitations. In this section, we therefore first discuss the extent to which altimetry data are applicable to our case. After that, the evidence for/against the existence of a Circum-Faroe Slope Current (CFSC) is evaluated. Then, we discuss the re-circulation of the Southern Faroe Current and the typical flow pattern. Finally, we discuss the implications of our results for our understanding of the poleward flow of warm water between Iceland and Scotland.

**4.1 The applicability of altimetry data**

The region under investigation is far from coastal boundaries that may destroy the quality of altimetry data, but there are other potential problems such as ageostrophic current components and the fact that altimetry is not a continuous measurement but a discrete set of overpasses and therefore prone to sampling errors.

For the SLA data, our main argument for their applicability is the fairly good correspondence with the independently measured ADCP velocities even though the ADCPs measure at a geographical point and well below the surface whereas the SLA difference between two altimetry points is related to the surface current averaged over the interval between the points. For this to be possible, velocity variations must be fairly barotropic, which seems to be the case for most of our ADCP sites (Table 2), but the velocity variations must also have a consistent horizontal structure. If, for instance, a current of narrow width sweeps back and forth past the ADCP site as it varies in strength, but remains between two altimetry points, then we would not expect a good correspondence between ADCP velocities and altimetry-derived velocities even with perfect altimetry data and geostrophy (Fig. S6b).

This point may be important for interpreting the values for the ADCP sites in Table 4. For site ZA over the Faroe slope, the altimetry explained about 70 % ($0.84^2$) of the variance in weekly averaged eastward velocity for two different deployments and the regression coefficients were almost identical (Table 4). With a distance, $L$, of a quarter of a degree in latitude between the two altimetry points z2 and z3, the theoretical value for $\alpha = g/(f \cdot L)$ is 2.8 s$^{-1}$ for surface velocity, that is much less than indicated by the regression analysis (Table 4). Together, the high correlation and high $\alpha$-value imply that the surface flow field between z2 and z3 must have a fairly consistent horizontal structure with the core consistently located close to site ZA (Fig. S6a). Also, the SLA data must be fairly good close to this site, especially when averaged over a week or longer.

At the other end of the Z-section, we find a similar picture with high correlations and high $\alpha$-values at site ZE over the Scottish slope. In the middle of the channel, ZB and ZC have lower correlation coefficients. One reason for that may be that the current core is more constrained laterally over the steep topography of the slopes than in mid-channel (Fig. S6). On the Z-section, which is the boundary between the Faroe-Shetland Channel and the Wyville Thomson Basin, altimetry therefore should represent variations in the slope currents fairly well on weekly time scales. On longer time scales the signal to noise ratio should increase and correspondence become better, which to some extent is supported in Table 4.

In the Faroe Bank Channel (sites FG, FB, and FC), the correlation coefficients in Table 4 are generally lower than on the Z-section. For sites FB and FC, this may partly be explained by the large depths at which ADCP velocities had to be sampled, but this is not the case for site FG. All three of these ADCP have different average velocities (Fig. 3, Table S2) and are not very highly correlated with one another (Table 3) when taking into account their proximity. We should therefore not expect a very consistent horizontal structure of the surface current (Fig. S6a). If that is the case, then the difference in SLA between f1 and f2 ($\Delta SLA_{FBC}$, Fig. 5) may be a better indicator of the surface flow through the Faroe Bank Channel than indicated by the correlation coefficients in Table 4. That might explain some of the differences between the Z-section and the F-section in Table 4, but must remain conjecture at this point.

In contrast to the SLA values, values for the Mean Dynamic Topography do not vary in time, but they depend upon an accurate determination of the geoid. By comparing Absolute Dynamic Topography values with in situ current measurements, Hansen et al. (2015) found that, north of the Faroes, the Mean Dynamic Topography values gave too weak and too broad currents indicating that the geoid may be too smooth on small scales. Since our region is similar, we therefore use Mean Dynamic Topography data cautiously and mainly for illustrative purposes.

## 4.2 The Circum-Faroe Slope Current

In the introduction, we raised the question, whether a part of the Southern Faroe Current continues through the Faroe Bank Channel and circulates the Faroes over the slope as a CFSC. For water from the Southern Faroe Current to continue into the Faroe Bank Channel, it has to pass through the Z-section, but the average velocity at 200 m depth through that section during the mooring experiment was eastward (Fig. 3) except at ZA where it was < 1 cm s$^{-1}$ indicating little westward Atlantic water transport through the section, especially when taking the bottom temperature into account (Fig. 4 and Fig. S2).

At ZA we also found a high correlation between eastward velocity at 200 m depth and the difference in altimetric SLA values between two grid points (Table 4), which allows this velocity component to be reproduced for the whole altimetry period (Fig. 5). Although variable, the average eastward velocity at 200 m depth at this site over the whole period 1993–2014 was close to zero.

Thus, it seems clear that on the average, there is not a substantial flow of Atlantic water from the Faroe-Shetland Channel to the Faroe Bank Channel through section Z off the shelf. In spite of this average condition, there are, however, extended periods with westward flow of Atlantic water at ZA (Fig. 4 and Fig. 5). If this water continues northwestwards through the Faroe Bank Channel, there might be a CFSC during these periods, but Fig. 5 does not indicate any clear relationship between surface flows through the Faroe Bank Channel (red curve) and at ZA (blue curve).

That does not, however, mean that there is no relationship between surface flows through the Faroe Bank Channel and the Faroe-Shetland Channel as demonstrated in Fig. 9. There, we have again used the difference in SLA between f1 and f2 ($\Delta SLA_{FBC}$) to represent the surface flow through the Faroe Bank Channel. Given the locations of the two altimetry points (Fig. 2a), positive values for $\Delta SLA_{FBC}$ indicate stronger than normal flow towards the Northwest.

In Fig. 9, values for $\Delta SLA_{FBC}$ have been correlated with simultaneous sea level anomalies in the whole region, where we use SLA* instead of SLA values to reduce the noise level (Fig. S7). The figure shows isolines of (negative) correlation coefficients following the slope on the Scottish side of the channel. The pattern is even clearer when we consider the regression coefficient (SLA* in all grid points regressed on $\Delta SLA_{FBC}$), which is plotted in Fig. 9b, but with the sign reversed for better illustration. The isolines for this coefficient may be interpreted as contour lines of the anomalous sea level associated with strong negative (i.e. southeastward) flow through the Faroe Bank Channel. Since the geostrophic surface current is parallel to contour lines of sea level, these isolines may also be interpreted as streamlines for the anomalous surface flow associated with this situation.

If a substantial part of the Southern Faroe Current in the surface were to continue through the Faroe Bank Channel, we might expect to see similar patterns as in Fig. 9, but the isolines should follow the slope on the Faroese side of the channel. Instead, they follow the Scottish side. If the surface water in the Faroe Bank Channel were to derive from the Faroe-Shetland Channel, they would, according to this, come from the Scottish slope region, but there the flow goes towards the Northeast, i.e. away from the Faroe Bank Channel. That is clear from the average velocity vectors at site SD and SE in Fig. 3 and this flow is very stable, especially at SE (Table S2). Of the 5235 days with velocity measurements at SE, only 268 days (5 %) had reversed flow direction. Thus, the water over the Scottish slope of the Faroe-Shetland Channel cannot feed the Atlantic water flow through the Faroe Bank Channel.

A more reasonable interpretation of the pattern in Fig. 9 is that negative (southeastward) surface flow through the Faroe Bank Channel joins the Atlantic inflow through the Faroe-Shetland Channel and flows northeastwards over the Scottish slope. In this interpretation, Fig. 9 illustrates variations in the strength of this flow. This only tells us about variations; not the average flow patterns. To the extent that we can rely on the Mean Dynamic Topography, the average surface flow pattern (Fig. S1) has a southeastward flow on the western side of the Faroe Bank Channel. Part of this continues into the Faroe-Shetland Channel, while the remainder re-circulates in the Faroe Bank Channel. This recirculation is enhanced when $\Delta SLA_{FBC}$ is highly positive (Fig. S8a). When $\Delta SLA_{FBC}$ is highly negative, a strong southeastward flow through the Faroe Bank Channel joins with inflow from the West to flow into the Faroe-Shetland Channel (Fig. S8b).

These conclusions are supported by the surface drifter tracks (Fig. 8), which show that all drifters, passing through the Faroe Bank Channel, came from the West or Northwest (Fig. S5). In addition to the surface drifters, Rossby et al. (2009) report the tracks of several RAFOS floats that were deployed west of the Iceland-Faroe Ridge in 2004 and 2005 and drifted at nominal depth 200 m. Five of these floats passed south of the Faroes and joined the pole-ward Atlantic inflow through the Faroe-Shetland Channel. One float circled the Faroes in a clockwise direction, passing from the Faroe-Shetland Channel into the Faroe Bank Channel (Fig. 5 in Rossby et al., 2009). At its entry into the Faroe-Shetland Channel, it was located over the outer Faroe Shelf, but loss of tracking signals prevents us from knowing whether it stayed over shallow areas or passed into deeper waters along its path through the Faroese Channels.

For the uppermost part of the Atlantic layer, we thus find no evidence for a continuous flow from the Faroese slope region into the Faroe Bank Channel. This seems to be the case for both the average and for instantaneous conditions

although there is the possibility of some water from the Southern Faroe Current occasionally being released into the Wyville Thomson Basin in the form of eddies (Fig. 7b) from where it may pass through the Faroe Bank Channel.

The deep parts of the Atlantic layer in the Faroe-Shetland Channel are colder and less saline (Fig. S3) and they may have a greater tendency to follow the overflow water towards the Faroe Bank Channel, but from Table 2, the variations in eastward velocity at ZA are highly correlated from 57 m to 257 m depth. It therefore seems likely that they will follow the surface flow pattern. If some of this water continues into the Faroe Bank Channel, it will be much colder and less saline than the Atlantic water that crosses the Iceland-Faroe Ridge (Hansen et al., 2003, 2015; Larsen et al., 2012). All the evidence, thus, points against a substantial CFSC.

## 4.3 The typical flow pattern

The existence of the Southern Faroe Current (Fig. 1) is well established and we see it at ADCP site EB, as well as at SX (Fig. 3), although less stable (Table S2). If not continuing through the Faroe Bank Channel, it has to re-circulate, at least in the upper parts of the Atlantic layer. When the Atlantic water flow through the whole of the off-shelf part of the Z-section is eastward, the Southern Faroe Current has to re-circulate east of the section, but what happens during the periods with strong westward flow through the Z-section over the Faroe slope (Fig. 4 and Fig. 5)?

In Fig. 10, we have plotted sea level anomalies in periods when the reproduced velocity at 200 m depth at ZA was westward, and again we use SLA* values to reduce the noise level. As before, the isolines of the SLA* field may be interpreted as streamlines for the anomalous surface velocity associated with this situation. The pattern shown in Fig. 10 indicates that the Atlantic water passing westward past ZA re-circulates due west of the Z-section, at least in the surface. This is also the case for the one surface drifter in the data set that passed westwards through the Z-section (blue track in Fig. 8b) and consistent with the hydrographic conditions during the May 2012 cruise of MRV Scotia (Fig. 6d, f).

The counterclockwise rotation east of the Z-section in Fig. 10 might give the impression that the surface water just circulates around a point in the middle of the Z-section, but note that this is just the anomalous sea level slope pattern associated with westward flow at ZA. To this should be added the average pattern, which counteracts the anomaly pattern east of the Z-section (black arrow at site SC in Fig. 10 based on 5235 days of ADCP measurements).

Thus, a complete counterclockwise circulation, as indicated in Fig. 10, may not necessarily always be associated with westward flow at ZA. It does, however, occur occasionally, as clearly demonstrated by the two blue drifter tracks in Fig. 8a and Fig. 8b. This type of feature was indicated already by Helland-Hansen and Nansen (1909, their Fig. 29). It has been described in terms of eddies that are generated in the frontal zone north of the Faroes and carried into the Faroe-Shetland Channel by the Southern Faroe Current (Hansen and Meincke, 1979; Dooley and Meincke, 1981) and Chafik (2012) has linked this to the strength of the NAO (North Atlantic Oscillation) index. Sherwin et al. (1999) have suggested more local generation at least for some of the eddies, and Sherwin et al. (2006) suggested baroclinic instability within the Faroe-Shetland Channel as a generating mechanism.

Without going into a more detailed discussion of the origin of these mesoscale motions, we note that the circulation sketch in Fig. 1 may give an overly simplified picture of the circulation within the Faroe-Shetland Channel. As demonstrated by the drifter tracks (Fig. 8), cross-channel flow may occur in both directions. The overall picture is, however, consistent with a complete re-circulation in the sense that all the Atlantic water that enters the Faroe-Shetland Channel seems to leave it by passing into the Norwegian Sea, mainly over the Scottish side of the channel.

Based on our results and the discussion in the previous section, Fig. 11 illustrates the typical Atlantic water flow through the Faroese Channels. The circulation is largely consistent with Fig. 1, although a bit more complicated with re-circulation of water from the Faroe Bank Channel in the Wyville Thomson Basin, a moving Midwater front, and cross-channel flow in both directions in the Faroe-Shetland Channel although crossing from the Scottish to the Faroese side of the Faroe-Shetland Channel (dashed arrows in Fig. 11) seems less persistent than crossing in the opposite direction.

### 4.4 The flow across the southern tip of the Faroe Shelf

Our rejection of a substantial Circum-Faroe Slope Current does not necessarily exclude substantial westward flow across the southernmost tip of the Faroe Shelf. According to Larsen et al. (2008), there is a persistent clockwise circulation of Faroe Shelf water that is mainly driven by tidal rectification. In that study, it is not clear, however, how much of that circulation manages to cross the narrow southern tip of the Faroe Shelf. Site CS (Fig. 3) at a bottom depth of 140 m on the eastern flank of the shelf shows flow along the bottom contours; not across, but site ZQ farther south and slightly deeper (Table S2) has an average westward velocity close to 5 cm s$^{-1}$ throughout the water column.

Unfortunately, we do not have any current measurements between sites ZQ and ZA, but if we assume a linear variation of the vertically averaged flow between these two sites, we can calculate the average volume transport through the section based on the ADCP measurements. From the average ADCP velocities at site ZB above the 5 °C isotherm, this can be extended southwards from ZA to the point where the average westward flow is zero. In this way, we estimate the total volume transport of Atlantic water south of the Faroes to be 0.4 Sv on average during the deployment periods. The westward velocity at ZA was well correlated with the SLA difference between altimetry points z2 and z3 (Table 4) and the interval between these two points extends from ZA more than halfway to ZQ. This motivates adding an additional 1 cm s$^{-1}$ westward flow (Sect. 3.2) through the whole cross-section to give a total transport of 0.5 Sv westward flow of Atlantic water south of the Faroes on average from 1 Jan 1993 to 31 Dec 2014.

The tidal currents are very strong over the southern tip of the Faroe Shelf (Larsen et al., 2008) and it may be argued that this rough estimate misses a tidally-driven flow that we have not observed. To check that, we have consulted the high-resolution ocean-model published by Rasmussen et al. (2014). This model has a typical horizontal grid size of 1 km over detailed topography and represents the tidally rectified current fairly well at those locations where it has been observed. According to this model, the southward-flowing water on the eastern flank of the Faroe Shelf turns eastwards; not westwards, and the model does not indicate any westward flow across the southern tip of the shelf on average over the 10-year period (2000 – 2009) of its run (Fig. 5 in Rasmussen et al., 2014). In addition to this, we have argued (Sect. 4.3, Fig. 10)

that much of the water flowing westward over the slope re-circulates in the Wyville Thomson Basin and returns to the Faroe-Shetland Channel. We therefore conclude that the 0.5 Sv, estimated above, is an upper limit for a "Circum-Faroe Current" including flows over both the shelf and the slope.

### 4.5 Implications

The results of our study have important implications when considering potential environmental impacts from offshore oil and gas activity in the Faroe-Shetland Channel. So far, this activity has mainly been focused over the Scottish side of the channel, but exploration has been carried out over the Faroese side, as well. From our results, potential harmful releases from this activity that enter and remain in the Atlantic layer in the Faroe-Shetland Channel are not likely to continue into the Faroe Bank Channel. Rather, they will cross to the Scottish side of the channel and continue towards the Norwegian Sea.

There are also implications for the oceanic heat transport towards the Arctic. The Atlantic inflow between Iceland and Scotland accounts for more than ¾ of the total inflow to the Arctic Mediterranean (Nordic Seas and Arctic Ocean) in terms of volume transport (Hansen et al., 2015). Since it is the warmest inflow branch, its transport of heat towards the Arctic will be even more dominant. A number of studies have emphasized the importance of this heat transport for Arctic sea ice (Årthun et al., 2012; Onarheim et al., 2014; Rippeth et al., 2015; Zhang, 2015; Polyakov et al., 2017) and it affects the
regional climate and living conditions for fish stocks of significant economic potential (Mork et al., 2014; Utne et al., 2012).

         In recent literature, observational estimates of volume and heat transport of the Atlantic inflow to the Norwegian Sea have mainly been obtained by two different methods. The studies by Berx et al. (2013) and Hansen et al. (2015) were based on long-term ADCP moorings and regular CTD cruises combined with satellite altimetry. The study by Rossby and Flagg (2012) was based on a ferry-mounted ADCP combined with historical hydrographic data. These two different methods give
different transport values (Table 5).

         Part of the disagreement may be explained by differences in the methods. As emphasized by Rossby and Flagg (2012), their estimate includes the circulation on the Faroe Shelf, which the other studies do not. North of the Faroes, this helps give better agreement. For the Faroe-Shetland Channel, this will also tend to give better agreement, but Rossby and Flagg (2012) also include flow in the opposite direction (their Fig. 2) over the Scottish shelf, not included in the Berx et al. (2013) study.
Thus, the overall effect of this for the Faroe-Shetland Channel will be small.

         The discrepancy is reduced if we assume that the closed circulation around the Faroes includes a flow of 1 Sv off the shelf in addition to the shelf circulation. Such a "tidally-driven circum-Faroe boundary current" (Fig. S9b) was assumed by Rossby and Flagg (2012). From our results, this assumption is not valid, which leads to large discrepancies between the two methods whether considering the full periods of the studies (Table 5) or a common period (Fig. S9).

One may argue that the important values are the total transports (Faroe-Shetland Channel + Iceland-Faroe Ridge), which do not disagree that much although the ferry-based estimates give 15 % less volume transport between Iceland and Scotland than estimated by Berx et al. (2013) and Hansen et al. (2015). These totals are, however, formed by summing the

contributions from the Faroe-Shetland Channel and the Iceland-Faroe Ridge. If the individual contributions disagree between the different methods, the relative agreement between totals must therefore be considered a fortuitous accident.

Another methodological difference is in the definition of Atlantic water. Rossby and Flagg (2012) use the criterion $\sigma_\theta <$ 27.8 kg m$^{-3}$, whereas Berx et al. (2013) use 5 °C in the Faroe-Shetland Channel and Hansen et al. (2015) use 4 °C for the colder water of the Faroe Current. North of Faroes, this may help bring the two estimates close to agreement but in the Faroe-Shetland Channel, a sensitivity analysis (Fig. 7 in Berx et al., 2013) indicates that the effect is marginal.

Finally, temporal transport variations might explain some of the discrepancies and Childers et al. (2014), using an updated data set from the ferry, have invoked that explanation to explain differences in transport from two different ferry tracks (Fig. 2a). During the period, in which Rossby and Flagg (2012) collected their ADCP measurements in the Faroe-Shetland Channel (March 2008 to March 2011), the average inflow according to Berx et al. (2013) was slightly below average (their Fig. 10), but it was still around 2.5 Sv (Fig. S9a). This period was furthermore mainly characterized by eastward flow at ZA (gray area in Fig. 5). Even if there should sometimes be a CFSC, it is not likely to have been during this period. Thus, the Rossby and Flagg (2012) measurements have to be interpreted without a CFSC, which implies an Atlantic inflow of 1.5 Sv through the Faroe-Shetland Channel including the flow over the Scottish shelf (Fig. S9c).

The two different methods for estimating transport of Atlantic water into the Norwegian Sea seem to agree fairly well north of the Faroes but for the Faroe-Shetland Channel, our rejection of a CFSC implies that the difference in volume transport estimated by the two methods is considerably larger than the uncertainty intervals quoted by the two studies. This needs to be resolved if we are to claim confidence in values for the transport of warm water towards the Arctic. To that end, a series of workshops has been initiated, the first of which was held in Tórshavn in January 2017 (Larsen et al., 2017).

**Data availability**

The observational data acquired by the Faroe Marine Research Institute are available online at www.envofar.fo while the data acquired by Marine Scotland Science are available online at the British Oceanographic Data Centre (www.bodc.ac.uk).

**Competing interests**

The authors declare that they have no conflict of interest.

*Acknowledgements.* Funding for the in situ measurements, data analysis, and manuscript preparation has been obtained from the European Framework Programs under grant agreement No. GA212643 (THOR) and grant agreement No. 308299 (NACLIM), and from the European Union's Horizon 2020 research and innovation program under grant agreement No 727852 (Blue-Action). We thank two anonymous referees for very constructive comments.

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

Table 1. Correlation coefficients between weekly averaged along-channel (eastward) velocities at ≈200 m depth (for ZQ–ZA, ≈120 m) at five ADCP sites on the Z-section during deployments in 2011–2012 and 2013–2014. "Weeks" indicates the number of values for each correlation.

| ADCP pair: | ZQ–ZA | ZA–ZB | ZA–ZC | ZA–ZE | ZB–ZC | ZB–ZE | ZC–ZE |
|---|---|---|---|---|---|---|---|
| Corr. coeff.: | +0.20 | +0.58*** | -0.79*** | -0.67* | -0.59*** | -0.70** | +0.68* |
| Weeks: | 36 | 84 | 36 | 20 | 36 | 20 | 20 |

**Table 2.** Correlations between weekly averaged along-channel velocities at the shallowest measured level and a level 200 m deeper for four ADCP sites, using the longest ranging deployment at each site of at least 11 months duration. "Weeks" indicates the number of values for each correlation. The last row lists the regression coefficient, a, in the equation: $U_{deep} = a \cdot U_{shallow} + b$.

| ADCP site: | ZA | FC | FB | FG |
|---|---|---|---|---|
| Depths (m): | 57–257 | 275–475 | 250–450 | 122–322 |
| Weeks: | 48 | 48 | 49 | 51 |
| Corr. Coeff.: | 0.97*** | 0.86*** | 0.30 | 0.95*** |
| Regr. Coeff.: | 0.76±0.06 | 0.84±0.15 | 0.43±0.43 | 0.98±0.09 |

**Table 3.** Correlations between pairs of weekly averaged along-channel (towards 304°) velocities for the three ADCP sites in the Faroe Bank Channel at their reference depths, which are 398 m for FC, 343 m for FB, and 122 m for FG. "Weeks" indicates the number of values for each correlation.

| ADCP pair: | FB–FC | FB–FG | FC–FG |
|---|---|---|---|
| Weeks: | 481 | 49 | 49 |
| Corr. Coeff.: | 0.66*** | 0.74*** | 0.41** |

**Table 4.** Correlation (R) and regression ($\alpha$ and $\beta$) coefficients between eastward velocity at selected depths within the Atlantic layer at 7 ADCP sites and SLA difference between pairs of altimetry points. Correlation coefficients are shown for daily ($R_1$), weekly ($R_7$), and 15-day ($R_{15}$) averaged data. The regression coefficients are based on weekly averaged data and according to Eq. (1). Where the ADCP site was close to midways between two neighbouring altimetry points, those points were used. Otherwise, we used the altimetry pair that gave the highest correlation for weekly averaged data. The two bottom lines are based on several deployments.

| ADCP | Depth | Period | Altim. | Duration | $R_1$ | $R_7$ | $R_{15}$ | $\alpha$ | $\beta$ |
|------|-------|--------|--------|----------|-------|-------|----------|----------|---------|
|      | m     |        | points | Days     |       |       |          | $s^{-1}$ | $cm\ s^{-1}$ |
| ZA | 197 | 2011–2012 | z3–z2 | 341 | 0.74*** | 0.83*** | 0.83*** | 5.3±1.1 | -1.4±2.9 |
| ZA | 197 | 2013–2014 | z3–z2 | 343 | 0.74*** | 0.84*** | 0.85*** | 5.2±1.0 | -1.3±2.8 |
| ZB | 200 | 2011–2012 | z4–z3 | 256 | 0.48** | 0.61** | 0.57* | 5.1±2.4 | 4.3±3.9 |
| ZB | 200 | 2013–2014 | z4–z3 | 343 | 0.42** | 0.51** | 0.57* | 3.2±1.6 | 9.0±3.2 |
| ZC | 200 | 2011–2012 | z5–z4 | 256 | 0.65** | 0.73** | 0.78* | 6.4±2.2 | 16.1±4.6 |
| ZE | 209 | 2011–2012 | z6–z5 | 145 | 0.62** | 0.74* | 0.92** | 5.0±2.5 | 10.9±5.1 |
| ZE | 268 | 2013–2014 | z6–z5 | 322 | 0.71*** | 0.83*** | 0.92*** | 4.4±0.9 | 10.9±2.1 |
| FG | 122 | 2008–2009 | f2–f1 | 364 | 0.42** | 0.58** | 0.71** | 4.0±1.7 | 9.2±3.3 |
| FB | 398 | 1995–2015 | f2–f1 | 6763 | 0.35*** | 0.51*** | 0.61*** | 2.7±0.3 | 2.2±0.5 |
| FC | 343 | 2001–2015 | f2–f1 | 4076 | 0.24*** | 0.41*** | 0.54*** | 1.7±0.3 | -10.3±0.6 |

**Table 5.** Volume transport (in Sv) of Atlantic water flow through the Faroe-Shetland Channel (FSC) and across the Iceland-Faroe Ridge (IFR) as determined by different methods. The estimates by Rossby and Flagg (2012) and by Childers et al. (2014) include a closed circulation on the Faroe Shelf (0.6 Sv) and flow on the Scottish Shelf, which are not included in the estimates by Berx et al. (2013) and Hansen et al. (2015).

| Studies | Period | FSC | IFR | FSC+IFR |
|---|---|---|---|---|
| Berx et al. (2013) + Hansen et al. (2015) | 1995–2009 | 2.7 | 3.8 | 6.5 |
| Rossby and Flagg (2012) | 2008–2011 | 0.9 | 4.6 | 5.5 |
| Childers et al. (2014) | 2008–2012 | 1.5 | 4.6 | 6.1 |

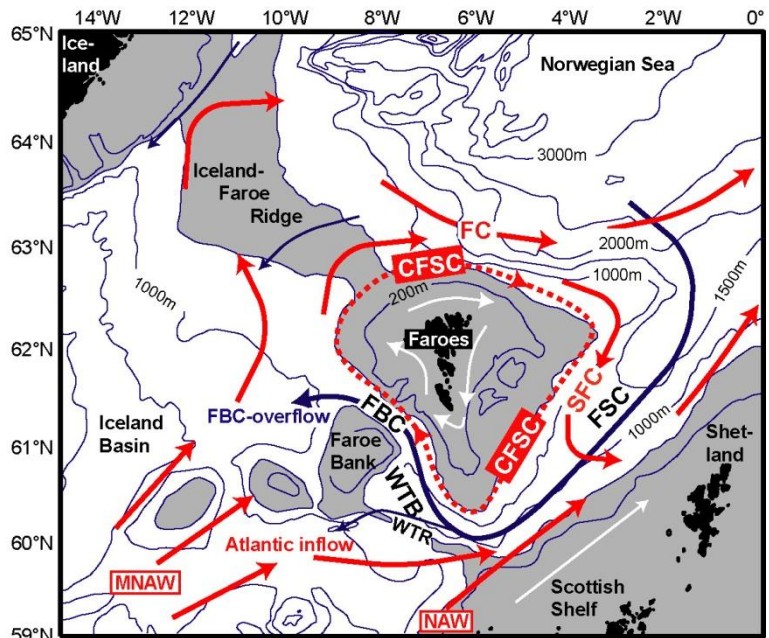

**Figure 1.** Bathymetry of the Faroese Channels, including schematic arrows of flow pathways. Gray areas are shallower than 500 m. Continuous arrows indicate well-established flows of: cold overflow water at depth (blue arrows), warm Atlantic inflow as North Atlantic Water (NAW) and as Modified North Atlantic Water (MNAW) in the upper layers off the shelves (red arrows), and shelf water (white arrows). Dashed red arrows indicate the hypothetical Circum-Faroe Slope Current (CFSC). FSC = Faroe-Shetland Channel, FBC = Faroe Bank Channel, WTB = Wyville Thomson Basin, WTR = Wyville Thomson Ridge, FC = Faroe Current, SFC = Southern Faroe Current.

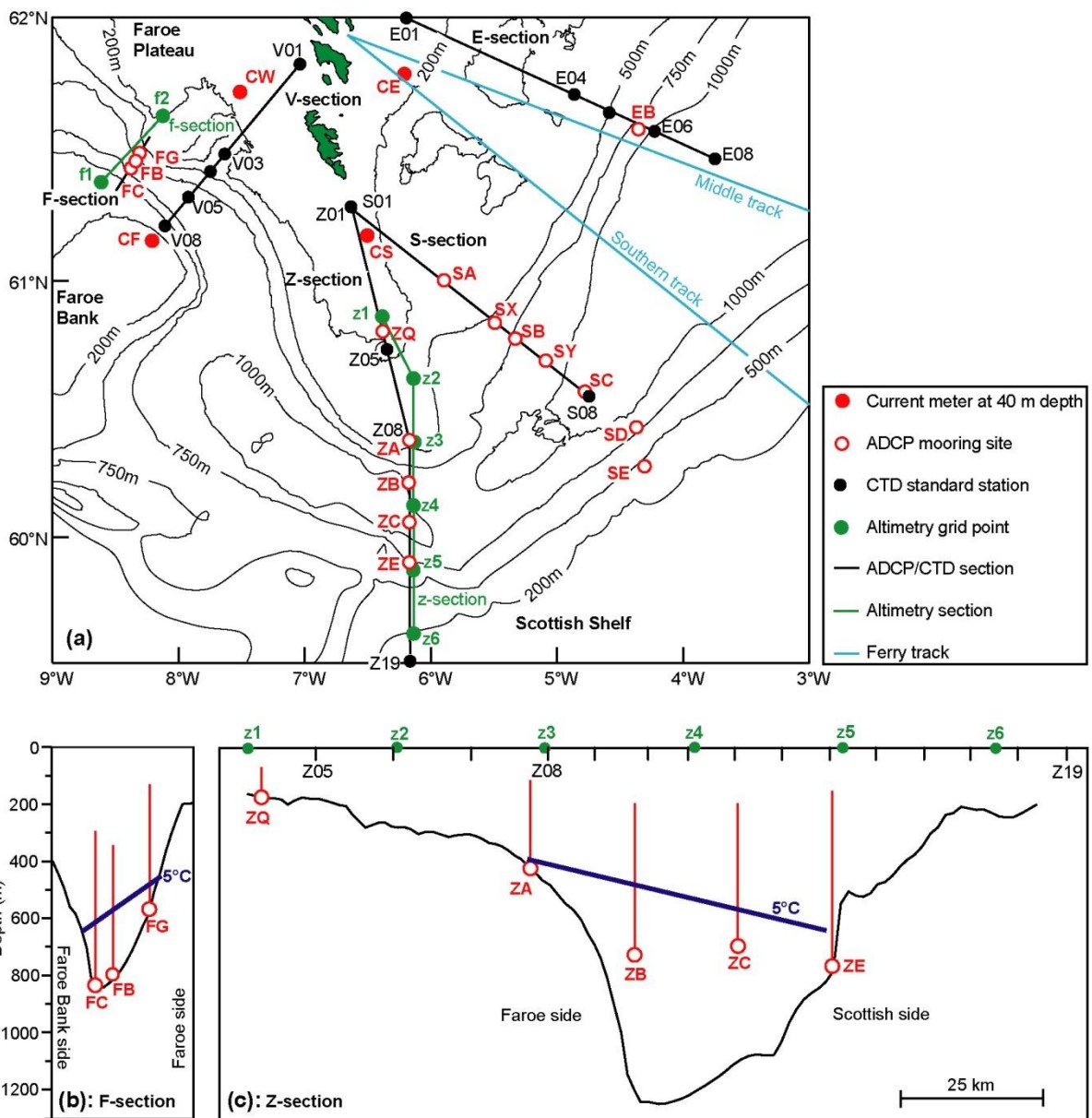

**Figure 2.** Locations of observational data: map view **(a)**, and cross-section views of **(b)** the F-section, and **(c)** the Z-section. Vertical red lines on the cross-sections indicate range of 100% good daily averaged ADCP data. Blue lines indicate the average depths of the 5 °C isotherms on the sections (from Hansen and Østerhus, 2007 and Hansen et al., 2013). ADCP sites and CTD stations are labeled by uppercase letters, whereas altimetry grid points are labeled by lowercase letters. The two cross-sections are drawn in the same vertical and horizontal scales.

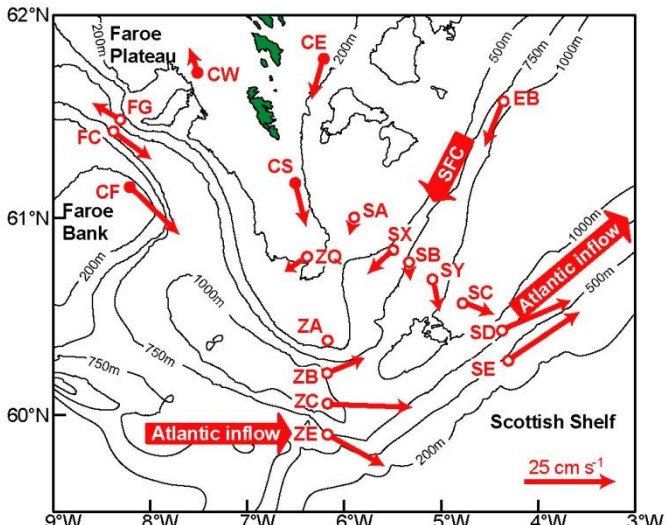

**Figure 3.** Atlantic water circulation observed by single-point (filled circles) and profiling (open circles) current meters. Red arrows indicate average velocity vectors at 40 m depth for the single-point moorings (Table S1) and at selected (see Table S2) depths within the Atlantic layer for the ADCP sites. Velocity scale is indicated in the lower right corner. At site ZA, the velocity was too weak for an arrow to be visible in the chosen scale.

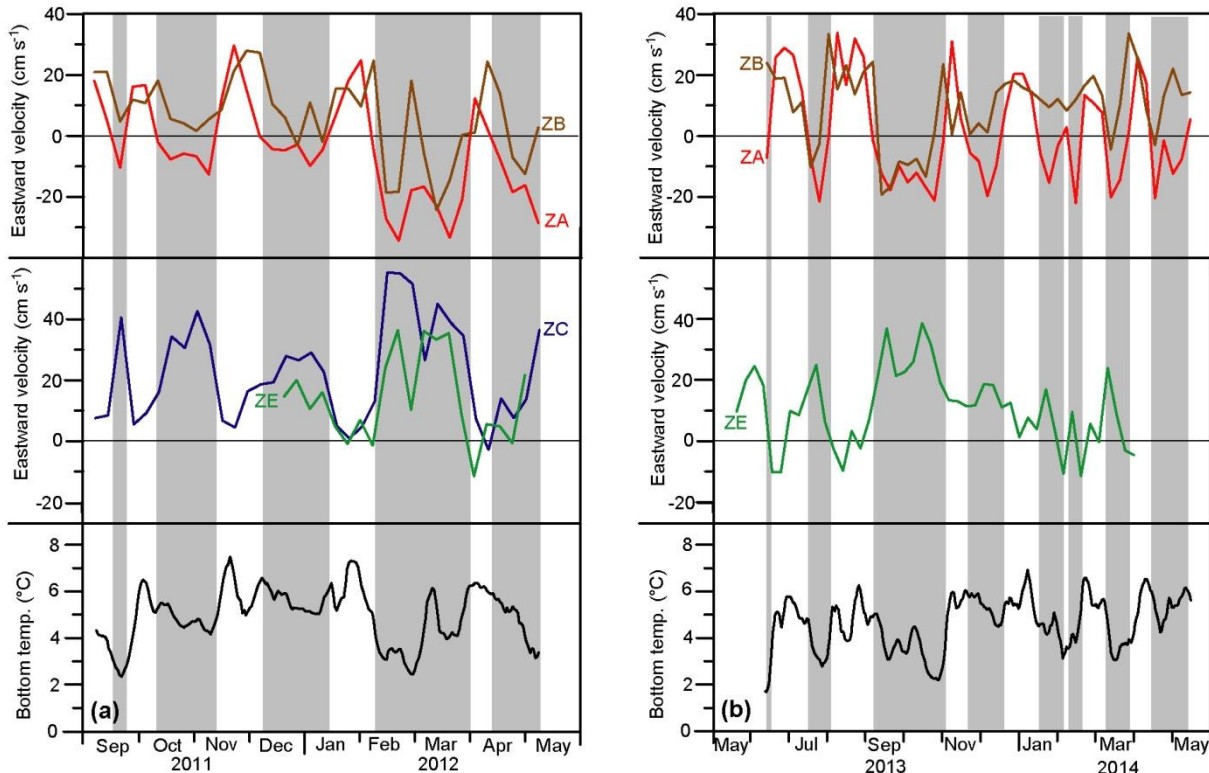

**Figure 4.** The top and middle panels show weekly averaged eastward velocity at ≈200 m depth at four ADCP moorings on the Z-section **(a)** from September 2011 to May 2012 and **(b)** from May 2013 to May 2014. Top panels are for the Faroese and middle panels for the Scottish side of the channel. Gray areas indicate periods when the velocity at ZA is westward. Due to limited ADCP range, the curve for ZE in **(b)** is at 268 m depth. The bottom panels show weekly averaged bottom temperature at ZA for the same periods.

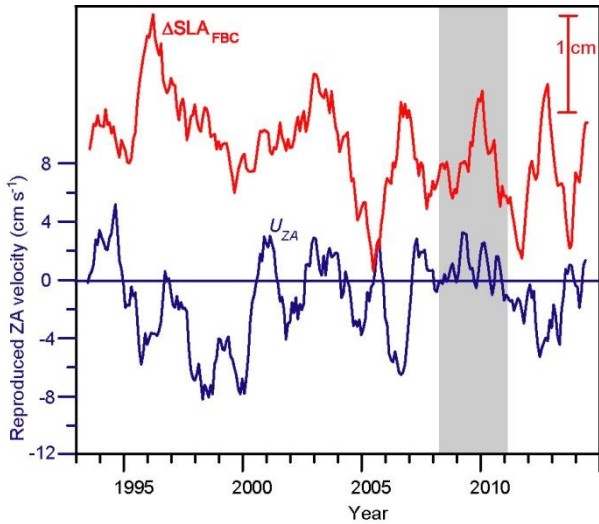

**Figure 5.** Annually (13 months) filtered values of: reproduced eastward velocity at 200 m at ZA ($U_{ZA}$, blue) and difference in SLA-values between f1 and f2 (f2-f1: $\Delta SLA_{FBC}$, red). The gray area indicates the period of ferry-based ADCP measurements by Rossby and Flagg (2012).

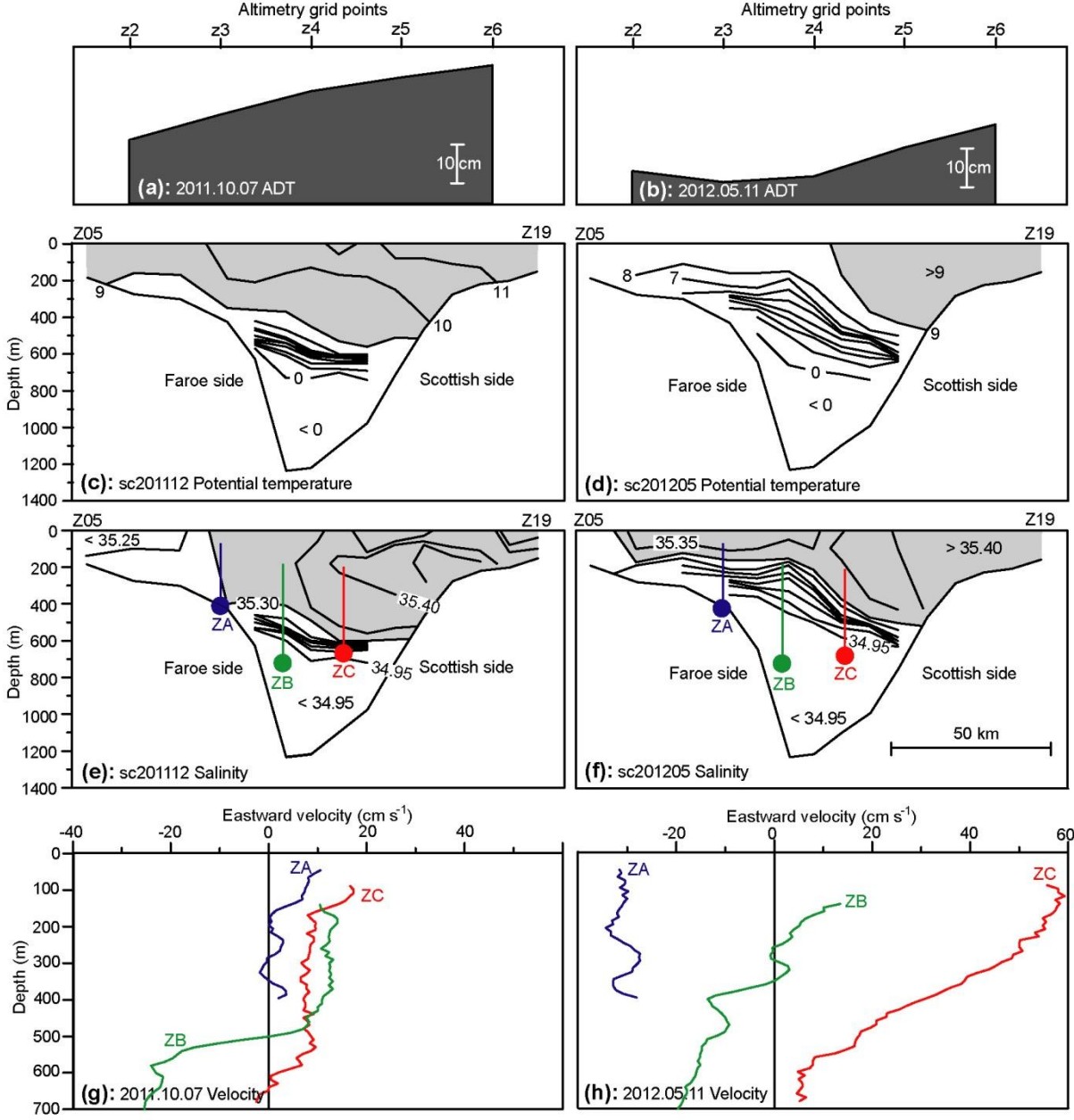

**Figure 6.** Conditions on the Z-section during two different cruises by MRV Scotia: Cruise sc201112 on 7–8 Oct. 2011 (**a**, **c**, **e**, **g**), and cruise sc201205 on 10–12 May 2012 (**b**, **d**, **f**, **h**). (**a**, **b**) Absolute Dynamic Topography (ADT). (**c**, **d**) Potential temperature distribution (in °C, contouring interval: 1 °C). Gray areas indicate water warmer than 9 °C. (**e**, **f**) Salinity distribution (contouring interval: 0.05). Gray areas indicate water more saline than 35.30. (**g**, **h**) Eastward component of ADCP velocity profiles (ADCP locations indicated on panels **e** and **f**). Panels **a**, **b**, **g**, and **h** are for dates in the middle of each cruise but would not look significantly different for other days within the cruise periods.

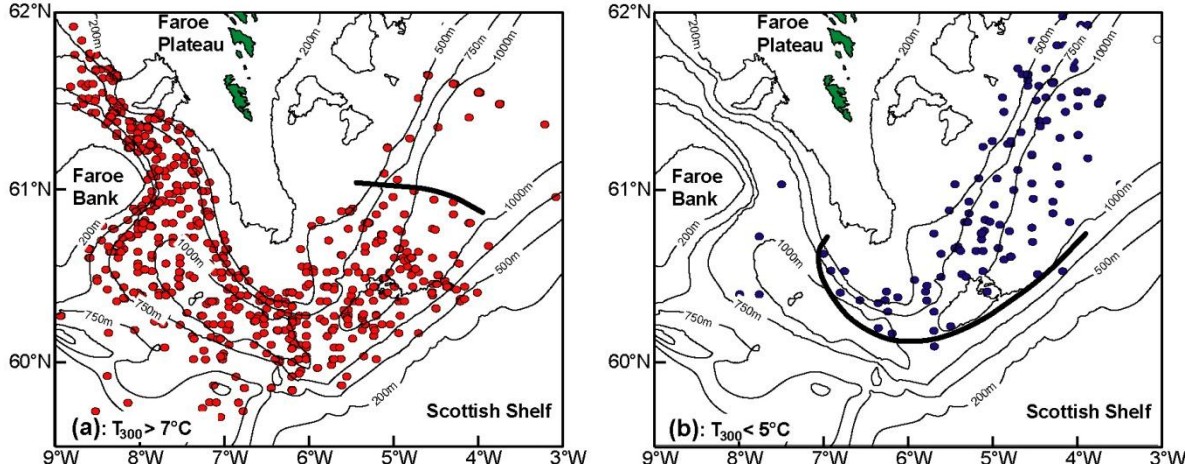

**Figure 7.** Temperature at 300 m depth. **(a)** Red dots show locations with temperature at 300 m depth higher than 7 °C. Black curve indicates the approximated eastern border. **(b)** Blue dots show locations with temperature at 300 m depth lower than 5 °C. Black curve indicates the approximated western border. Based on 2905 CTD profiles obtained by R/V J. Chr. Svabo or by R/V Magnus Heinason 1976–2015 at locations with bottom depth at least 400 m in the region.

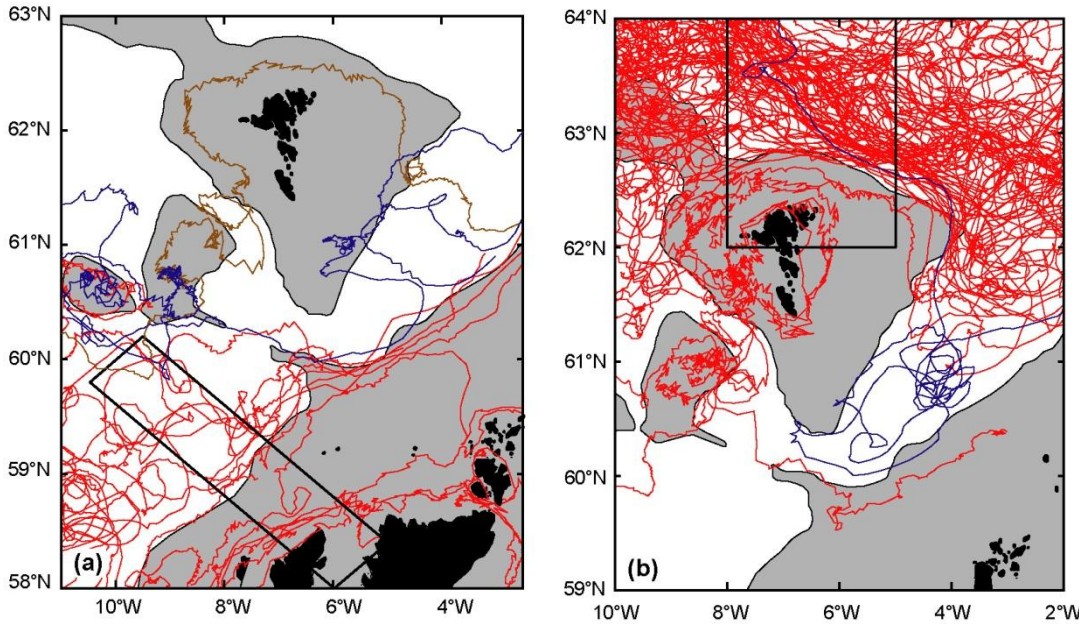

**Figure 8.** Tracks of 104 surface drifters passing through or close to the Faroese Channels. **(a)** Tracks of 16 surface drifters that entered the polygon shown southwest of the Wyville Thomson Ridge. **(b)** Tracks of 89 drifters that entered the rectangle shown north of the Faroes (extended northwards to 66° N).

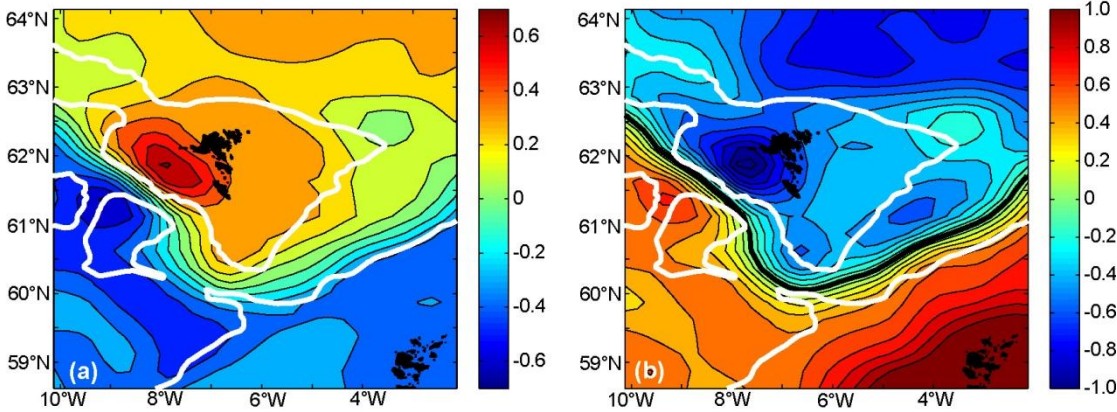

**Figure 9.** Relationships between Atlantic water transport through the Faroe Bank Channel and altimetry values in the region. **(a)** Correlation coefficient between $\Delta SLA_{FBC}$ and SLA* in all grid points of the region. **(b)** Negative value of the regression coefficient (slope) of SLA* in all grid points regressed on $\Delta SLA_{FBC}$, where the zero value is indicated by a thick black line. The thick white lines indicate the 500 m bottom contour. Based on weekly averages.

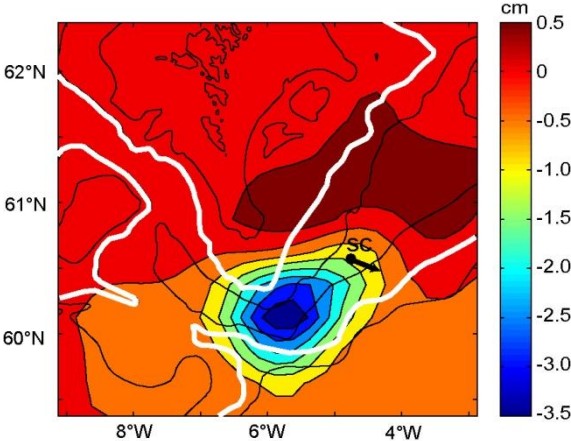

**Figure 10.** Average SLA* values in all grid points of the region in periods when the reproduced (Eq. (1)) velocity at 200 m depth at ZA is westwards. The average velocity direction measured by ADCP at 200 m depth at site SC is shown by a black arrow (copied from Fig. 3).

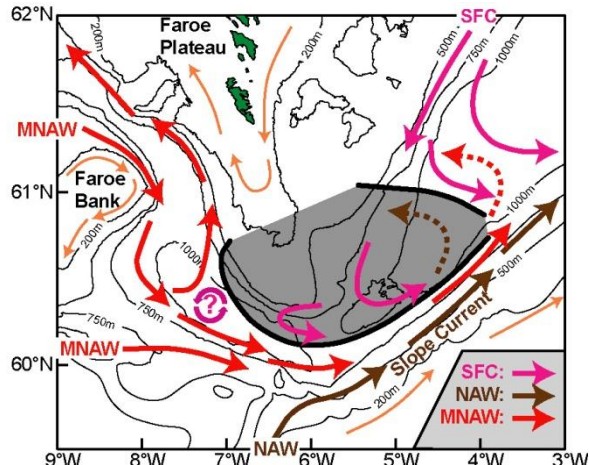

**Figure 11.** Proposed scheme for the typical passage of Atlantic water through the Faroese Channels. Thin orange arrows indicate flow paths on the shelves and Faroe Bank. Thick coloured arrows show suggested flow paths for three different water masses off the shelves. Gray area indicates the region within which the Midwater Front is typically located (from Fig. 7). The circled question mark indicates possible eddies from the Southern Faroe Current released into the Wyville Thomson Basin.