# Peer review of "Atlantic water flow through the Faroese Channels"

_Ocean Science, 2017_

## Referee Comment (RC1) · Anonymous Referee #1 · 24 Jul 2017

General comments

This manuscript discusses new evidence showing that a previously hypothesized current around the Faroes in fact does not exist, or at least is not a consistent feature of the circulation. This is important information, which I believe should be distributed to the scientific community. Although the paper is structured clearly, readability is somewhat hampered by the myriad of acronyms employed by the authors. Furthermore, although I think their evidence from situ observations is convincing, I do have some concerns regarding the satellite data analysis as outlined below.

Specific comments

1. The satellite altimetry analysis glosses over major issues such as the ageostrophic

component(s) of the currents near topographic boundaries (which are not captured by altimetry), general issues with lack of reliability of altimetry data near coastal boundaries, and the fact that altimetry is not a continuous measurement but a discrete set of overpasses and therefore prone to sampling errors in a highly variable area such as the Faroese Channels system. If the authors wish to include altimetry data in this manuscript, a thorough discussion of uncertainties in the analysis should be included.
2. Disagreement between different heat flux estimates through a section is to be expected as is it impossible to calculate (see e.g. Schauer and Besczcynska-Möller, Ocean Sci., 5, 487-494, 2009). Unless I misunderstood and the heat flux estimates from previous work were based on closed contours, it makes little sense to compare them. Volume fluxes of course are fine and should be compared.
* * *

---

## Referee Comment (RC2) · Anonymous Referee #2 · 18 Aug 2017

The manuscript analyses observational oceanographic data from Faroese waters in order to clarify a question regarding the local circulation of Atlantic Water. Therefore, the recordings of moored current meters and ADCPs, CTD profiles, satellite altimetry and surface drifters from the region south of the Faroes including the Faroese shelf, the Faroe-Shetland Channel, the Faroe-Bank-Channel and the Wyville Thomson Basin are used. The temporal ranges of the observations reach from 1976-2015 (CTDs), 1993-2015 (satellite altimetry) up to 2011-2014 (ADCPs).

The question the authors try to answer is whether or not the Atlantic Water of the Faroe Current which flows eastwards north of the islands completely enters the Norwegian Sea. East of the Faroes a part of the Faroe Current branches off southwards into the Faroe-Shetland Channel. This water could either form a Circum-Faroe Current or recirculate northwards within the Faroe-Shetland Channel. The first case was proposed by Rossby and Flagg (2012) who analysed ADCP data recorded aboard the high-seas ferry M/F "Norröna" on her route between Iceland, the Faroes and Denmark. With this, Rossby and Flagg contradicted the currently accepted theory which is based on the second case.

Because the current meters, whose records are temporally extended with the help of the altimetry data, only show a weak westward transport south of the islands and because the hydrographic observations south-west of the Faroes do not point to a large influx of Faroe Current water the circulation scheme proposed by Rossby and Flagg is rejected.

The warm and saline Atlantic Water which crosses the Greenland-Iceland-Scotland-Ridge forms the upper branch of the Meridional Overturning Circulation of the North Atlantic, a process of great importance of the regional climate because of the involved oceanic heat transports. In order to evaluate their models climate researchers need realistic, observational based circulations schemes of that key region. But these schemes are still far from being clear, unambiguous and ensured. Often enough, ocean modellers have desperately screwed their settings in order to fit an established circulation scheme until this was finally replaced by something much more in agreement with the initial numerical solution. A look at the oceanographic literature about the circulation along that ridge shows a row of conflicting schemes and dramatic changes of the established ones which happened during the last decades. Even the schemes denoted as "well-documented" in this manuscript are perhaps rather "well-established" or "often used" by established authors. Hence, we are just beginning to answer questions which stem from the 19th century. Therefore, the manuscript is an important piece of oceanographic work and I recommend the publication with some minor corrections or clarifications.

General comments

Generally, the manuscript does not convince me that the transport of the South Faroe Current south of the Faroes is zero. The circled question mark in Figure 11 makes the impression of saying: "We found SFC water south-west of the islands (Fig. 7b) but this must not be caused by a current." Though, the current meter at station ZA shows a westward flow during most of the time, which becomes even more pronounced within the temporally expanded time series shown in Figure 5. Also the fact that in Figure 3 the ZA station symbol does not come with an arrow makes a somehow unfortunate impression. The reader may ask why there is no current meter station at the most interesting position between ZQ and ZA, and how this record would look like. What would be the consequences if it would show a westward component comparable to station SX and FG? My impression of the presented data is that there is a small, shallow part of the SFC which is not recirculating but joining the clockwise coastal circulation. Its volume transport is obviously smaller than assumed by Rossby and Flagg. But I don't understand the absoluteness this branch is negated by the authors. This should be made clearer or changed.

Rossby and Flagg mention the tidal forcing of the proposed circum-Faroer boundary current. Why this forcing is not discussed? Is it possible to obtain a 1.5 Sv residual current south of the Faroes? Aren't there any ocean models including the tides and assimilating the huge regional CTD data base, and perhaps even the current meter records, which could be used to examine the SFC dynamics? I do not demand an additional model study, but at least one sentence about the structure proposed by one of the state-of-art ocean models of that region would make sense.

The excessive and unnecessary usage of abbreviations – water masses, channels, basins, currents, instruments, data sets etc. – makes the manuscript hard to read. From my point of view, only SFC, CTD and ADCP are useful. It would be great if this could be reduced a bit.

Specific comments

Page 1, line 13 and 27: "water masses from the Atlantic" as far as I know the term "Atlantic" comprises the entire ocean including the Nordic Seas.

Page 2, line 18: "Hátun (2004) called this current the Southern Faroe Current (SFC)" Why did he call a current east of the Faroes "Southern"?

Page 3, line 15: "these measurements were not, however, made". I would change it to: these measurements were, however, not made . . .

Page 3, line 24: "Table S1" does not exist

Page 4, line 14: "For some applications..." If only spatial differences are used the subtraction of a constant value applied to the field should not have any effect.

Page 4, line 25: "Data on surface drifter tracks . . ." too brief description. State at least the temporal range and the number of drifters.

Page 5, line 25: "(e.g. SB – FG, Table S3)", the pair SB – FG does not exist in Table 3 (S3?)

Page 6, line 18: "do not appear very barotropic", change to "do not appear to be very barotropic"

Page 6, line 28: "has good data", what does it mean? Be more precise.

Page 7, line 9: "From the altimetry data set, we have selected 6 altimetry points" the second "altimetry" is redundant.

Page 7, line 24: "for <alpha> (g/(f L))", change to "for <alpha> = g/(f L)",

---

## Author Comment (AC1) · 20 Sep 2017

Below, please find our responses to referee comments. The revised manuscript and supplement are added as one merged supplementary pdf file where non-trivial changes are in red.

Anonymous Referee #1

Comment 1.1: Although the paper is structured clearly, readability is somewhat hampered by the myriad of acronyms employed by the authors.

Response: We have followed the recommendation by both reviewers to reduce the number of acronyms/abbreviations and have only kept "ADCP", "CTD", and "SLA", which all are widely used in the literature, "SLA*", which is a derivative of "SLA", and

"CFSC", which is the main topic of this manuscript. The other acronyms (abbreviations) have been replaced by the full names. In contrast to other changes, these have not been marked in red in the revised manuscript.

Comment 1.2: Furthermore, although I think their evidence from situ observations is convincing, I do have some concerns regarding the satellite data analysis as outlined below. Specific comments 1. The satellite altimetry analysis glosses over major issues such as the ageostrophic component(s) of the currents near topographic boundaries (which are not captured by altimetry), general issues with lack of reliability of altimetry data near coastal boundaries, and the fact that altimetry is not a continuous measurement but a discrete set of overpasses and therefore prone to sampling errors in a highly variable area such as the Faroese Channels system. If the authors wish to include altimetry data in this manuscript, a thorough discussion of uncertainties in the analysis should be included.

Response: We agree with the referee that the use of altimetry data to infer current velocity, a priori, is fraught with assumptions and uncertainties. Nevertheless, we feel that the altimetry data help answer some of the questions addressed in the manuscript and we feel that the good correlations between ADCP velocities and SLA differences support the applicability of altimetry data for some problems, at least. But, this has probably not been sufficiently clear. So, we have followed the suggestion by Referee 1 to present a more thorough discussion in the new Sect. 4.1 in the revised manuscript where we have collected the more distributed discussion in the original manuscript as well as adding new information including an expanded Table 4 and a new Fig. S6 in the Supplement.

Comment 1.3: Disagreement between different heat flux estimates through a section is to be expected as is it impossible to calculate (see e.g. Schauer and Besczcynska-Möller, Ocean Sci., 5, 487-494, 2009). Unless I misunderstood and the heat flux estimates from previous work were based on closed contours, it makes little sense to compare them. Volume fluxes of course are fine and should be compared.

Response: We believe that heat transport values may be more realistic in our region than in the Fram Strait studied by Schauer and Besczcynska-Möller (2009), especially when referenced to a temperature (0°C) that should be close to the average temperature of all the outflow branches from the Arctic Mediterranean, but much colder than the Atlantic water in our region. Also we find that oceanic heat transport in this region is an important concept, which should not be ignored although it may be hard to measure. But, we agree that there is an (unavoidable ?) ambiguity involved in calculating heat transport of individual branches following open contours. We have therefore deleted all heat transport values (TW) from Table 5 and elsewhere in the manuscript and now only refer to heat transport more qualitatively.

Anonymous Referee #2

Comment 2.1: Even the schemes denoted as "well-documented" in this manuscript are perhaps rather "well-established" or "often used" by established authors.

Response: "well-documented" has been replaced by "well-established".

Comment 2.2: Generally, the manuscript does not convince me that the transport of the South Faroe Current south of the Faroes is zero. The circled question mark in Figure 11 makes the impression of saying: "We found SFC water south-west of the islands (Fig. 7b) but this must not be caused by a current." Though, the current meter at station ZA shows a westward flow during most of the time, which becomes even more pronounced within the temporally expanded time series shown in Figure 5. The reader may ask why there is no current meter station at the most interesting position between ZQ and ZA, and how this record would look like. What would be the consequences if it would show a westward component comparable to station SX and FG? My impression of the presented data is that there is a small, shallow part of the SFC which is not recirculating but joining the clockwise coastal circulation. Its volume transport is obviously smaller than assumed by Rossby and Flagg. But I don't understand the absoluteness this branch is negated by the authors. This should be made clearer or changed.
Response: It was not our intention to claim that there is no westward flow at all south of the Faroes including the shelf region, but we realize that the original manuscript may well be interpreted in that way. To amend that, we have added a new section to the discussion, Sect. 4.4. The current measurements on the Z-section were not originally designed for the purpose of this manuscript, which is one reason that there was no measurement between ZQ and ZA. In the new Sect. 4.4, we use the available observational evidence and results from a model to make a rough estimate of volume transport and argue that the combined shelf/slope volume transport is not likely to exceed 0.5 Sv on average. We have also added a sentence on this to the abstract.

Comment 2.3: Also the fact that in Figure 3 the ZA station symbol does not come with an arrow makes a somehow unfortunate impression.

Response: We have added the text: "At site ZA, the velocity was too weak for an arrow to be visible in the chosen scale" to the caption of Figure 3.

Comment 2.4: Rossby and Flagg mention the tidal forcing of the proposed circum-Faroer boundary current. Why this forcing is not discussed? Is it possible to obtain a 1.5 Sv residual current south of the Faroes? Aren't there any ocean models including the tides and assimilating the huge regional CTD data base, and perhaps even the current meter records, which could be used to examine the SFC dynamics? I do not demand an additional model study, but at least one sentence about the structure proposed by one of the state-of-art ocean models of that region would make sense.

Response: Thank you for this suggestion. For a model to capture tidal forcing, we assume that it has to have a high resolution and realistic topography. We only know of one published model for the Faroe Shelf that fulfils these criteria and have now included results from that model in the new Sect. 4.4.

Comment 2.5: The excessive and unnecessary usage of abbreviations – water masses, channels, basins, currents, instruments, data sets etc. – makes the manuscript hard to read. From my point of view, only SFC, CTD and ADCP are useful.

It would be great if this could be reduced a bit.

Response: See our response to Comment 1.1.

Specific comments Comment 2.6: Page 1, line 13 and 27: "water masses from the Atlantic" as far as I know the term "Atlantic" comprises the entire ocean including the Nordic Seas.

Response: This has been corrected and different phrasing used for these two cases and elsewhere in the manuscript.

Comment 2.7: Page 2, line 18: "Hátun (2004) called this current the Southern Faroe Current (SFC)" Why did he call a current east of the Faroes "Southern"?

Response: Most likely, the reason was that the SFC is south of (but not east of) the Faroe Current. We agree that the name may not be optimal, but do not think it wise to introduce a new name.

Comment 2.8: Page 3, line 15: "these measurements were not, however, made". I would change it to: these measurements were, however, not made ...

Response: Has been done.

Comment 2.9: Page 3, line 24: "Table S1" does not exist

Response: It appears that the reviewer has not realized that Table S1 is in the supplement. To emphasize this, we have added the text "in the supplement" here (but not for other references to the supplement).

Comment 2.10: Page 4, line 14: "For some applications..." If only spatial differences are used the subtraction of a constant value applied to the field should not have any effect.

Response: Yes, our original phrasing was probably misleading. We have now removed the sentence: "since only spatial differences are required to find geostrophic currents".

To help make the motivation for introducing SLA* clearer, we have now added a reference to Fig. S7 to Sect. 2.3.

Comment 2.11: Page 4, line 25: "Data on surface drifter tracks ..." too brief description. State at least the temporal range and the number of drifters.

Response: Has been done.

Comment 2.12: Page 5, line 25: "(e.g. SB – FG, Table S3)", the pair SB – FG does not exist in Table 3 (S3?)

Response:See response to Comment 2.9, above.

Comment 2.13: Page 6, line 18: "do not appear very barotropic", change to "do not appear to be very barotropic"

Response: Has been done.

Comment 2.14: Page 6, line 28: "has good data", what does it mean? Be more precise.

Response: Has been replaced by: "for which only a few days were error-flagged".

Comment 2.15: Page 7, line 9: "From the altimetry data set, we have selected 6 altimetry points" the second "altimetry" is redundant.

Response: "altimetry points" has been replaced by "grid points".

Comment 2.16: Page 7, line 24: "for <alpha> (g/(f L))", change to "for <alpha> = g/(f L)"

Response: Has been done.

Please also note the supplement to this comment:
https://www.ocean-sci-discuss.net/os-2017-47/os-2017-47-AC1-supplement.pdf

**Supplement:**

[revised manuscript text omitted]

**Figure S6.** Two conceptual examples to illustrate the relationship between surface current velocity measured by current meter (e.g., ADCP) or by altimetry. Both panels show eastward surface current (u) plotted against the y-coordinate (northward) at two different times. $A_1$ and $A_2$ are two altimetry grid points between which the horizontally averaged velocity $u_A(t)$ can be derived assuming geostrophy. O is a point at which the surface current $u_O(t)$ is measured by current meter. In **(a)**, we assume that the horizontal structure of the surface velocity is perfectly consistent so that the spatial and temporal variation may be separated: $u(y,t) = \varphi(y) \cdot \psi(t)$. In this case, both $u_A(t)$ and $u_O(t)$ will be proportional to $\psi(t)$ and hence proportional to one another. If all measurements and geostrophic balance are perfect, the correlation coefficient between $u_A(t)$ and $u_O(t)$ will be 1. The regression coefficient α in Eq. (1) in the main manuscript will be higher the closer O is to the core of the current. At the core, it will be well above the theoretical value. With noisy data and/or ageostrophic flow, the correlation coefficient will be less than 1, but will be highest close to the core. In **(b)**, we assume a narrow surface current, for which the strength does not vary with time, but the current moves back and forth laterally. In this case, $u_A(t)$ will be constant whereas $u_O(t)$ varies. With perfect measurements and geostrophy, the correlation coefficient between $u_A(t)$ and $u_O(t)$ will be 0.

[Figure]

**Figure S7.** The difference between using SLA and SLA*. **(a)** Correlation coefficient between $\Delta SLA_{FBC}$ and SLA at all grid points of the region. **(b)** Correlation coefficient between $\Delta SLA_{FBC}$ and SLA* at all grid points of the region.

[Figure]

**Figure S8. (a)** Average MDT+SLA* when $\Delta SLA_{FBC} \geq +1$ standard deviation. **(b)** Average MDT+SLA* when $\Delta SLA_{FBC} \leq -1$ standard deviation. MDT is Mean Dynamic Topography. The thick white lines indicate the 500 m bottom contour.

[Figure]

[Figure]

**Figure S9.** Atlantic inflow between Iceland and Scotland (thick red arrows) March 2008 to March 2011 in three recent studies using two different methods. **(a)** According to Berx et al. (2013) and Hansen et al. (2015). **(b, c)** According to Rossby and Flagg (2012) with **(b)** and without **(c)** a Circum-Faroe Slope Current (CFSC) of 1 Sv. Rossby and Flagg (2012) find a closed circulation on the Faroe Shelf of 0.6 Sv (white arrows), which is not included in the estimates by Berx et al. (2013) and Hansen et al. (2015). In addition, Rossby and Flagg (2012) assume a closed CFSC of 1 Sv (**b**), which implies an inflow to the Faroe-Shetland Channel from the West of 2.5 Sv in close agreement with Berx et al. (2013). Without the CFSC (**c**), the inflow from the West is reduced to 1.5 Sv including flow over the Scottish Shelf that is not included in Berx et al. (2013).